# The Effectiveness of the Super Skills for Life (SSL) Programme in Promoting Mental Wellbeing among Institutionalised Adolescents in Malaysia: An Interventional Study

**DOI:** 10.3390/ijerph19159324

**Published:** 2022-07-30

**Authors:** Kishwen Kanna Yoga Ratnam, Nik Daliana Nik Farid, Nur Asyikin Yakub, Maznah Dahlui

**Affiliations:** 1Institute for Public Health, National Institutes of Health, Shah Alam 40170, Malaysia; kishwen87@gmail.com; 2Centre of Population Health, Department of Social and Preventive Medicine, Faculty of Medicine, Universiti Malaya, Kuala Lumpur 50603, Malaysia; daliana@ummc.edu.my (N.D.N.F.); 3Psychology and Human Well-Being Centre, Faculty of Social Sciences and Humanities, National University of Malaysia, Bangi 43600, Malaysia; nurasyikin6666@ummc.edu.my

**Keywords:** cognitive behavioural therapy (CBT), super skills for life (SSL) programme, adolescents, detention settings, mental health

## Abstract

Background: Mental health issues have become more prevalent among institutionalised adolescents. Therefore an effective intervention programme is needed to improve their mental health. Objective: To evaluate the effectiveness of the Super Skills for Life (SSL) programme in improving the mental wellbeing of institutionalised adolescents and determine the factors associated with their mental wellbeing. Methods: A quasi-experimental study involving 80 female institutionalised adolescents divided into intervention and control groups was conducted. Intervention involved implementation of the SSL programme. The effectiveness of the programme was evaluated based on several outcome parameters. Results: Factors including age, number of family members, perceived social support and self-esteem had significant correlations with mental wellbeing of participants. The SSL programme significantly improved the anxiety and stress levels of participants. Conclusion: SSL programme exclusively improves the mental wellbeing in institutionalised adolescents.

## 1. Introduction

Mental health issues have become more prevalent among institutionalised adolescents compared to non-institutionalised adolescents over the years. Delinquency and mental health issues pose a substantial risk to public health comparable to other risks, such as non-communicable and infectious diseases, among institutionalised adolescents [1]. In particular, adolescent institutionalisation within the juvenile justice system is of concern as mental disorders are found to be very common among juveniles [2]. Worldwide, 70–100% of institutionalised adolescents had at least one mental disorder classified in the Diagnostic and Statistical Manual of Mental Disorders IV (DSM-IV) [3]. In Malaysia, 93% of detained juveniles experienced one or more diagnosable mental disorders, including anxiety and stress [4]. According to the National Health and Morbidity Survey, 20% of institutionalised adolescents aged 10–17 years old suffered from depression, while 10% attempted suicide due to depression [5]. It is found that institutionalisation had a negative impact on their mental wellbeing and personal protective skills, such as resilience. They tend to become more emotional and less resilient as a result of institutionalisation [6]. Besides, institutionalisation also has a significant effect on the adolescents’ interpersonal and social developmental domains. They tend to have impaired cognitive development and are unable to form emotional attachment with their caregivers, which results in low social support received from family and friends [7]. Moreover, institutionalised adolescents also have higher risk of developing low self-esteem due to emotional abuse [8]. Due to all these mental health issues, institutionalised adolescents are unable to contribute to the community once they are released.

Despite the implementation of rehabilitative programmes in correctional institutions, the prevalence of mental health issues and recidivism is still rising among institutionalised adolescents [9]. For instance, the cases of juvenile offence in Malaysia had increased by 10.5% from 4833 cases in 2019 to 5342 cases in 2020 [10]. This creates a drive for the research of a more effective intervention programme to empower institutionalised adolescents so that they can reintegrate into the community after their release from custody. Cognitive behavioural therapy (CBT) based intervention, such as the Super Skills for Life (SSL) programme, may be an effective solution to this problem. The SSL programme is a targeted prevention programme with multi-component intervention strategies consisting of cognitive preparation, social skills and behavioural activation. It aims to cultivate socially competent adolescents by training them in social skills and by improving their mental wellbeing through the instillation of positive self-perceptions [11]. Evidence has shown the effectiveness of the SSL programme intervention in improving adolescents’ mental wellbeing and personal protective skills, such as self-esteem and coping skills [11,12,13,14]. For example, the SSL intervention has shown improvements to the symptoms of generalised anxiety and social phobia among institutionalised youth [12]. Furthermore, the self-esteem and coping skills of adolescents to deal with behavioural issues were also significantly improved after the SSL intervention [11,13,14].

However, there has not been any study in Malaysia available to explore the effectiveness of the SSL programme among institutionalised adolescents in a more robust manner. Therefore, this study aims to evaluate the effectiveness of the SSL programme in improving the mental wellbeing of institutionalised adolescents in the aspects of mental health status (depression, anxiety and stress) and personal protective skills (resilience, perceived social support and self-esteem). Moreover, factors associated with the mental wellbeing of institutionalised adolescents are also identified so that they can be mitigated to help with the success of the programme.

## 2. Materials and Methods

### 2.1. Study Design

This is a quasi-experimental study, which consists of a cross-sectional study on all eligible institutionalised adolescents divided into intervention and control groups, measuring various parameters of their psychological wellbeing pre-intervention; implementation of the SSL programme for the intervention group; post-intervention survey to evaluate immediate changes in observed outcomes; and follow-up assessment after two months for the measurement of sustained immediate effects.

### 2.2. Study Population

The study population included female participants residing in correctional institutions located in Peninsular Malaysia. All participants were female as the welfare institutions within the sampling pool comprised of only female correctional institutions. This limited sampling pool was due to the implementation of Movement Control Order because of the COVID-19 crisis, in which many detention centres restricted the implementation of programmes by external agencies due to health and safety concerns.

### 2.3. Inclusion and Exclusion Criteria

Detained female participants who were aged 12–22 years old and were able to read and understand either English or Malay were included in the study. Female participants who were quarantined for disciplinary problems or health reasons or have cognitive abnormalities were excluded from the study.

### 2.4. Sample Size

The sample size included in the study was calculated based on a previous study by Amiri, Heshmati, and Shad 2010 that investigated depression, anxiety and stress parameters among orphanage residents [15,16]. After calculation, the maximum number selected as the final sample size was *n* = 128 for both intervention and control groups.

### 2.5. Intervention Programme

Participants in the intervention group were arranged to participate in the SSL programme, whereas the control group would be excluded from the programme. The programme was implemented once a week over a period of two months. It was conducted by an experienced researcher, who had undergone intensive training by the author of the SSL programme, Professor Cecelia Essau, through a Training the Trainer (ToT) course organised in 2019 in Malaysia. The researcher was also appointed as a facilitator by the Malaysian Ministry of Youth and Sports to aid in the nationwide implementation of the SSL programme in secondary schools in 2019. The programme consisted of eight sessions, involving different activities as stated in Table 1.

Prior to the SSL programme, participants were subjected to a pre-test in the form of a questionnaire available in both English and Malay where the participants could choose from. Upon completing the SSL session, participants were requested to complete another survey to evaluate the immediate and follow-up effects of the intervention. Participants in the control group were also requested to complete the same pre-test, post immediate and follow-up test in the form of a questionnaire to assess the outcome parameters.

### 2.6. Data Collection and Analysis

The study extended over a period of five months from October 2020 to February 2021. The study was conducted in three randomly selected correctional institutions located in Selangor, Malacca and Terengganu, Malaysia. Participants in the intervention and control groups were from two geographically distinct institutions, with the intervention group from correctional institutions in Selangor and Malacca, while the control group was from a correctional institution in Terengganu. This was to prevent knowledge transfer and contamination of intervention between the two groups. Socio-demographic information of the participants was obtained and tabulated according to the participation forms submitted by the participants.

The effectiveness of the SSL programme was assessed based on the outcome parameters, including mental health status (depression, anxiety, stress and mental wellbeing), resilience, perceived social support, self-esteem and coping skills, which were evaluated according to the different scales featured in the questionnaire. Two scales were used to measure mental health status among the participants, namely the Depression, Anxiety and Stress Scale-21 (DASS-21) and the Warwick-Edinburgh Mental Wellbeing (WEMWBS) (mental wellbeing). In brief, DASS-21 comprises three subscales with 7 items that measure each of the depression, anxiety, and stress outcome with a 4-point Likert scale, ranging from 0 (never) to 3 (mostly) based on what they had experienced over the past week. DASS-21 has a reported reliability scores of 0.95, 0.85 and 0.87, respectively, for Depression, Anxiety and Stress domains.

Additionally, measurement of mental health status was also achieved with the WEMWBS, a validated scale consisting of 14 items in the form of a 5-point Likert scale, ranging from 1 (none of the time) to 5 (all the time), covering both hedonic and eudaimonic aspects of mental health, including positive affect (feelings of optimism, cheerfulness, relaxation), satisfying interpersonal relationships and positive functioning (energy, clear thinking, self-acceptance, personal development, competence and autonomy). It displayed a high reliability with Cronbach alpha of 0.90.

Furthermore, several personal factors were measured using a Resilience Scale (resilience), Multidimensional Scale of Perceived Social Support (MSPSS) (perceived social support), Rosenberg Scale of Self-Esteem (RSES) (self-esteem) and Brief-COPE (coping skills). In particular, the Resilience Scale is made up of 25 items which can be rated on a 7-point Likert scale, ranging from 1 (strongly disagree) to 7 (strongly agree) on statements about resilience. It has been shown to have high validity and reliability in diverse settings with alpha coefficient measurements of an internal consistency reliability of between 0.87 and 0.95.

The MSPSS, on the other hand, comprises a total of 12 questions with 4 questions exploring each of the different sources of social support, namely family, friends and a significant other. Each item in MSPSS is measured with 7-point Likert scale, ranging from 1 (strongly disagree) to 7 (strongly agree). It has displayed good internal consistency with Cronbach alpha value of 0.89.

For self-esteem measurement, RSES is a ten-item Likert scale with items answered on a four-point scale, from strongly agree to strongly disagree. With a Cronbach alpha value of 0.8, it uses a scale of 0–40 where a score less than 15 may indicate a problematic low self- esteem. Finally, Brief-COPE is an abbreviated version of the COPE (Coping Orientation to Problems Experienced) Inventory, a self-administered questionnaire developed to assess a broad range of coping responses. The instrument consists of 28 items that measure 14 factors of 2 items each, which correspond to a Likert scale ranging from 0 = I have not been doing this at all to 3 = I have been doing this a lot. This study used a translated and validated Malay version of the Brief COPE with a total Cronbach alpha value of 0.83.

Data obtained from the pre- and post-intervention and a 2 month follow-up assessment were analysed using the Statistical Package for Social Science (SPSS) desktop version 22.0 software (SPSS Inc., Chicago, IL, USA), focusing on intention to treat (ITT) calculations to evaluate the effectiveness of the SSL programme. The pre-post effect sizes were also calculated and compared. Pearson correlation analysis was used in inferential statistics to determine the contribution of each factor (socio-demographic and personal factors) and in predicting mental health status. Statistical difference was considered significant if the *p*-value was less than 0.05. Meanwhile, the mean difference of outcome parameters (depression, anxiety, stress, mental wellbeing, resilience, perceived social support, self-esteem and coping skills) between the intervention and control group for the three intervention phases (pre, post and 2 month follow-up) was assessed using a Bonferroni test. Assumptions including normality, homogeneity of variance and co-variance were verified before running the test. The effect size was also measured using partial eta squared, whereby 0.01 was considered small effect size, 0.06 was considered moderate effect size while 0.14 was considered large effect size.

### 2.7. Ethical Considerations

Ethics approval was obtained from the University of Malaya Research Ethics Committee (UMREC) with a reference number of TNC2/UMREC–817 prior to the commencement of the study. Written consent was obtained from all participants. For participants aged 18 and below, an additional informed consent was obtained from their respective legal guardians. The study was conducted in full compliance with the laws and regulations of the Malaysia Data Protection Act 1998. All information pertaining to participants’ mental wellbeing were kept confidential. Disclosure of any personal issues was dealt with confidentiality.

## 3. Results

### 3.1. Socio-Demographic Profile of Participants

A total of 80 participants took part in the study with 80 responses collected for analysis. Six responses were lost in follow-up due to absenteeism of the participants during the time of data collection. The flowchart of the study are shown in Figure 1.

The socio-demographic characteristics of the participants were recorded in Table 2. The mean age for the participants was 18.44 years old. A majority of participants were in the age range of 16–18, with 39.1% in the intervention group and 67.6% in the control group. According to Table 2, significant differences were observed between the groups for categories, including age (*p*-value = 0.005), number of siblings (*p*-value = 0.011), child order number in family (*p*-value = 0.017), number of family members living together in one household (*p*-value ≤ 0.001), parents’ marital status (*p*-value = 0.001) and duration in institution (*p*-value ≤ 0.001).

### 3.2. Mental Health Status, Resilience, Perceived Social Support, Self-Esteem and Coping Skills of Participants during Pre-Intervention Phase

The overall baseline outcome parameters of the included participants were illustrated in Table 3. Over 70% of the participants reported some level of mental health issues of varying degrees ranging from mild to extremely severe condition. Meanwhile, 57.5% of the participants showed low resilience level. For the aspect of perceived social support, 66.3% of the participants reported high levels of social support. In terms of self-esteem, the majority of the participants (78.8%) reported low self-esteem. All participants are considered to have high coping skills.

The responses of the control and intervention participants and their mental health status in terms of depression, anxiety and stress, resilience, perceived social support and self-esteem during the pre-intervention phase were compared and tabulated in Table 4. Homogeneity of the pre-test score between the two groups was established with comparable mean total scores in mental health status (depression, anxiety, stress and mental wellbeing), resilience, perceived social support and self-esteem. A significant difference was observed in the baseline coping skills, whereby the intervention group has higher coping skills compared to the control group (*p* = 0.049).

### 3.3. Factors Associated with Mental Health Status of Participants

From Table 5, it was found that both socio-demographic factors and personal factors correlated with the dependent variables: mental health status (depression, anxiety and stress), mental wellbeing, resilience, perceived social support and self-esteem. Age had a significant negative correlation with depression (r = −0.222; *p*-value < 0.05), indicating younger adolescents had a greater tendency of experiencing depression. The number of family members living in one household was found to be significantly correlated with depression (r = 0.261; *p*-value < 0.05), anxiety (r = 0.258; *p*-value < 0.05), perceived social support (r = 0.264; *p*-value < 0.05) and self-esteem (r = −0.250; *p*-value < 0.05). Among these variables, the number of residents living in one household had a significant negative correlation with self-esteem. This indicated that low self-esteem was influenced by increasing the number of family members in the household. Meanwhile, perceived social support had a significant negative correlation with stress (r = −0.261; *p*-value < 0.05), indicating stress was associated with low social support provided to the participants. Correlation analysis also discovered that self-esteem was significantly correlated with the participants’ mental health status, including depression (r = 0.351; *p*-value < 0.01), anxiety (r = 0.244; *p*-value < 0.01) and stress (r = 0.501; *p*-value < 0.01).

### 3.4. The Effectiveness of the Super Skills for Life (SSL) Programme

The effects of the SSL programme on various outcome parameters were evaluated using the repeated measure ANOVA. Based on the results shown in Table 6, significant improvement with SSL compared to control over time (pre- and post-intervention and 2 month follow up) was observed for anxiety (*p* = 0.012) and stress (*p* ≤ 0.001) outcome parameters. Interestingly, a significant improvement of depression outcome was observed when comparing between groups (*p* = 0.001) and time points (*p* = 0.040) separately, despite no effect being seen with the intervention over time. Time effect was also observed in mental wellbeing (*p* = 0.047) while group effect can be seen in perceived social support (*p* = 0.038), self-esteem (*p* ≤ 0.001) and coping skills (*p* ≤ 0.001) outcome parameters.

Considering the quasi-experimental study design and the variability found in some of the baseline socio-demographic and personal factors, 3-way interaction ANOVA was conducted to adjust the influence of these factors to the efficacy of SSL in improving anxiety and stress. As depicted by Table 7, age, number of siblings, child order number in family, number of family members living together in one household, parents’ marital status, duration of institutionalization and baseline perceived social support, did not influence the efficacy of SSL on anxiety and stress. In terms of baseline self-esteem, this personal factor seems to only affect stress (*p* = 0.009) and not anxiety.

## 4. Discussion

The current study evaluates the effectiveness of the SSL programme in improving the mental wellbeing of institutionalised adolescents in terms of mental health status (depression, anxiety and stress) and personal protective skills (resilience, perceived social support and self-esteem). Considering the risk of knowledge transfer and contamination, the study population for this study was derived from different institutions for each of the experimental groups. However, homogeneity of the baseline characteristics among the participants could not be established when there were significant differences on the results of the socio-demographic profile of the participants in terms of age, number of siblings, child order number in family, number of family members in one household, parents’ marital status and incarceration duration found.

In terms of outcome measure parameters, the variation observed with the socio-demographic characteristics were not apparent with participants’ mental health status of interest (depression, anxiety and stress, resilience, perceived social support and self-esteem). Nevertheless, study results before the intervention showed that over 70% of them experienced some forms of mental health issues of varying degrees of seriousness (Table 3). This finding corresponds with a previous systematic review encompassing 47 studies from 19 countries involving 28,033 male and 4754 female adolescents from juvenile correctional institutions. The research reported about 76.5% of female adolescents suffered from mental disorders, such as major depression and attention-deficit/hyperactivity disorder (ADHD) [1].

In addition, a total of 57.5% reported low scores in resilience before the intervention, while in contrast, a total of 66.3% of participants reported high scores in perceived social support prior to the intervention programme (Table 3). According to a study, it was found that female adolescents who were institutionalised due to delinquent behaviour developed pseudo-family relationships among themselves, where they often shared their feelings and belongings, thus forging a sense of camaraderie. Additionally, perceived social support in a closed community would need a collective behavioural change among a sizeable group of participants in order to generate an organic sustained perceived peer support. This would take practice and time before such genuine feelings of peer support were formed and felt [17]. This would explain how and where participants in this study received social support in the institution setting. However, there is no significant difference between two institutions in terms of mental health status. Besides, significant difference among social support between the two groups may be meaningless as the mean social support demonstrated by all the participants were considered in the range of high perceived social support.

Furthermore, the heterogeneity of the baseline characteristics of the participants motivates the correlation analysis between the socio-demographic characteristics and the outcome measured at baseline. The results showed that age was negatively correlated with depression (Table 4), which indicated that younger adolescents were more likely to experience depression compared to older adolescents. This finding contradicts a previous study, which reported that the estimated prevalence of depression among adolescents significantly increased with age, from 4% at age 13 to 19% at age 16 [18]. However, the targeted subjects of the study were adolescents who were not institutionalised. This current study, on the other hand, targeted institutionalised adolescents. Thus, the difference in findings may be due to the difference in circumstances of targeted subjects.

Next, the number of family members in one household was found to be correlated with depression, anxiety, perceived social support and self-esteem. Increasing number of residents in one household was found to have significantly increased the level of depression and anxiety as well as lowering the self-esteem of the participants. Although the study has not shown the association of the number of family members with depression, anxiety and self-esteem, family discord was identified to be one of the contributing factors for the occurrence of depression and anxiety among adolescents [19]. So, it is possible that increasing number of family members may invoke unpleasant competition among the children for the parents’ approval. In addition, larger families may have a difficulty with supporting their children due to limited resources, which then may lower the self-esteem and cause depression and anxiety in the children [20]. Nonetheless, its positive correlation with perceived social support is reasonable and desirable as this will be helpful for the participants to cope with their circumstances in confinement.

As for personal protective factors, low perceived social support and self-esteem were significantly correlated with depression, anxiety and stress. This finding corresponds with a previous study, which reported low self-esteem to be ranked as a modestly specific and highly feasible risk factor, while lack of social support was ranked as a modestly feasible risk factor for mental health issues in adolescents, such as depression, anxiety and stress [21]. Although the socio-demographic factors, such as age and number of family members, are difficult to manipulate, it is possible to improve the mental wellbeing of institutionalised adolescents by improving personal factors, such as perceived social support and self-esteem. This is possible through the intervention of the SSL programme.

Due to the heterogeneity of the participants, it is important to consider the confounding factors that may affect the analysis of the results. Significant improvement was observed for anxiety levels among the participants after the SSL intervention (Table 5). Similar findings are also noted in a study conducted among the Spanish youth who were at risk of developing anxiety disorders. The study reported a significant drop in symptoms of generalised anxiety and social phobia among the youth after the SSL intervention [12].

Next, stress emerged as another outcome that is positively impacted by SSL intervention among all three types of mental health status (Table 5). Apart from the acute characteristic of stress, another possible reason could be that the participants had understood and practiced the stress-relieving techniques that were taught during the sessions in SSL programme. Besides, they had even adopted a positive outlook towards their circumstances, which is the intended purpose of the SSL intervention.

Meanwhile, the level of depression showed some positive changes as compared with anxiety and stress, albeit insignificant when interaction of both time and intervention were considered (Table 5). This is probably due to the nature of depression, which requires more in-depth and targeted intervention, or even a combination of psychotherapy and pharmacotherapy [22]. Nonetheless, this finding contradicts with those of another study, which showed significant improvement in the symptoms of depression among children and adolescents aged 9–14 from the SSL intervention group [23].

When considered collectively, the mental wellbeing of participants showed a significant improvement as their incarceration duration progressed (Table 5). However, the effect of SSL on mental wellbeing could not be established when considering both time and intervention effect.

As mentioned earlier, perceived social support and self-esteem were correlated with mental health status of the participants (Table 4). This means that the improvement in perceived social support and self-esteem will reduce depression, anxiety and stress of the participants, which in turn will improve their mental wellbeing. As shown in the study results, improvements observed in depression, anxiety, stress, perceived social support and self-esteem did lead to better mental wellbeing among participants in the intervention group (Table 5). Since the majority of the participants were already institutionalised for more than six months before the commencement of the intervention, such drastic improvement of mental wellbeing (anxiety and stress) after the relatively short 2 month intervention was mainly attributed to the SSL programme.

On the other hand, the level of perceived social support recorded significant improvement in the SSL intervention group compared to the control group. However, this may be due to the differences observed in the perceived social support during pre-intervention phase instead of an effect produced by the SSL. When interaction between time and intervention was analysed, no significant difference could be established. Nevertheless, the level of the perceived social support in both groups are already considered high and might not be a clinically significant choice.

Similarly, significant improvement was observed in the self-esteem of the participants from intervention groups following enrolment in the SSL programme (Table 5). This finding corresponds with a study conducted among school children, which showed significant improvement in self-esteem after the SSL intervention [14]. However, this difference might be influenced by the different baseline characteristics of the participants from different institutions.

A similar improvement was also observed in coping skills among the participants after the SSL intervention (Table 5). The result was again in-line with those reported in previous studies, which showed that the coping abilities of juveniles to deal with their behavioural issues were greatly improved in both short- and long-term ways following the SSL intervention [11,23]. Similarly, this difference might be influenced by the different baseline characteristics of the participants at pre-intervention as no significant improvement of coping skills was observed when investigating the effect of time.

Finally, the SSL intervention was found not to change the level of resilience in this study (Table 6). According to Table 7, there was no interaction between time intervention and age, number of siblings, number of family members, parents’ marital status, incarceration duration, social support and self-esteem in the level of anxiety. Similarly, there were also no interactions found between time intervention and age, number of siblings, number of family members, parents’ marital status, incarceration duration, social support and self-esteem in the level of stress.

Overall, the institutionalised adolescents enrolled in the SSL programme from the intervention group reported sustained positive mental wellbeing as compared to those from the control group who demonstrated a more stagnant and insignificant improvement in their mental wellbeing (Table 5). This finding is consistent with a previous study conducted among school children. Similar findings, such as improvements in mood, self-esteem and a reduction in psychological discomfort, were observed among adolescents after the SSL intervention [13]. This is because the programme was designed to build emotional resilience among youth, while at the same time increasing their ability to deal with stressful life events, especially drastic life changes [24].

Besides, several previous studies also showed the effectiveness of the SSL programme in lowering the symptoms of depression and anxiety, while increasing the efficiency to manage short- and long-term behavioural issues among juveniles [11,14]. This is evident in the present study, in which there was a significant improvement in anxiety and stress (Table 5). This indicates the effectiveness of the SSL programme, whereby the knowledge and skills transferred to the participants were well received, practiced and normalised in their daily life, which then resulted in behavioural changes.

One of the limitations is the study design. This quasi-experimental study design may increase the possibility of selection bias, which may then affect the study outcome. Therefore, instead of a quasi-experimental study design, a randomised controlled study design is recommended to eliminate possible selection bias through the randomisation of participants.

Another limitation is the small sampling parameter that includes only a specific group of participants, which is made up of only Malay female adolescents. This limitation prevents analysis based on gender and ethnicity. Thus, the results of this study does not represent a generalised picture on the mental wellbeing of adolescents in all detention centres in Malaysia. Hence, the sampling frame should be extended in future research to include both male and female participants from all major ethnicities for a better representation of adolescents’ mental wellbeing.

## 5. Conclusions

It is noted that over 70% of female institutionalised adolescents have mental health issues. Factors including age, number of family members in one household, perceived social support and self-esteem are significantly correlated with the mental wellbeing of the participants. It is proven that the SSL programme produces significant positive effects on institutionalised adolescents’ mental health by reducing levels of stress and anxiety. In conclusion, the SSL programme is a safe and effective intervention that can improve the mental wellbeing of institutionalised adolescents.

## Figures and Tables

**Figure 1 ijerph-19-09324-f001:**
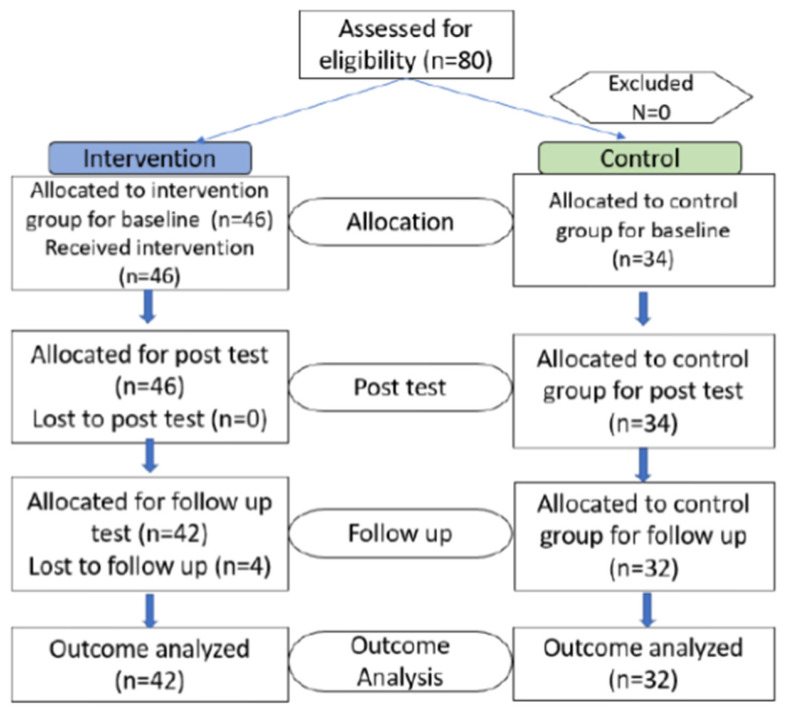
Participant flowchart of the study according to the CONSORT 2010 guidelines.

**Table 1 ijerph-19-09324-t001:** The SSL programme modules and activity description.

Sessions	Aims/Activities
Session 1	Introducing participants to Super Skills for Life (Adolescent version) (SSL-A) Discussing behaviours that promote a “healthy” lifestyle (i.e., eating healthy food, regular physical activities, enough sleep)
Session 2	Introducing participants to concept of self-esteem and discussing activities that can enhance self-esteem Discussing small steps needed to develop a particular skill
Session 3	Introducing participants to the concept of feelings and thoughts
Session 4	Introducing the concept of the link between thoughts, feelings and behaviour
Session 5	Learning about the impact of stress on the body and feelings Teaching participants specific relaxation strategies to deal with stress and anxiety
Session 6	Discussing the importance of having good relationships Learning specific skills needed to get along with people
Session 7	Learning how to use problem-solving steps to manage social problems
Session 8	Introducing the importance of having a sense of future Learning how to set short-, medium- and long-term goals

**Table 2 ijerph-19-09324-t002:** Socio-demographic characteristics of participants.

Categories	Frequency (n)/Percentage (%)		*p*-Value
	Intervention = 46	Control = 34	
**Age (years); mean = 18.44 years**			
13–15	3 (6.5)	3 (8.8)	0.005
16–18	18 (39.1)	23 (67.6)	
19–21	12 (26.1)	8 (23.5)	
≥22	13 (28.3)	-	
**Number of siblings**			
1	1 (2.2)	1 (2.9)	0.011
2	3 (6.5)	9 (26.5)	
3	4 (8.7)	8 (23.5)	
4	16 (34.8)	4 (11.8)	
≥5	22 (47.8)	12 (35.3)	
**Child order number in family**			
1st	12 (26.1)	21 (61.8)	0.017
2nd	13 (28.3)	3 (8.8)	
3rd	5 (10.9)	4 (11.8)	
4th	7 (15.2)	2 (5.9)	
5th	-	-	
**Number of family members living together in one household**			
1	-	2 (5.9)	<0.001
2	-	3 (8.8)	
3	3 (6.5)	4 (11.8)	
4	2 (4.3)	13 (38.2)	
≥5	41 (89.1)	12 (35.3)	
**Parents’ marital status**			
Married	32 (69.6)	13 (38.2)	0.001
Divorced	5 (10.9)	14 (41.2)	
Separated	-	4 (11.8)	
Widowed	8 (17.4)	3 (8.8)	
Passed away	1 (2.2)	-	
**Level of education**			
No formal education	-	1 (2.9)	0.435
Primary school	6 (13.0)	3 (8.8)	
Secondary school	40 (87.0)	30 (88.2)	
**Working experience**			
Yes	29 (63.0)	25 (73.5)	0.322
No	17 (37.0)	9 (26.5)	
**Household income**			
Below RM1500	10 (21.7)	6 (17.6)	0.272
RM1501–RM3000	6 (13.0)	5 (14.7)	
≥RM3001	7 (15.2)	1 (2.9)	
Do not know	23 (50.0)	22 (64.7)	
**Duration in institution (months)**			
≤5	4 (8.7)	9 (26.5)	<0.001
6–12	26 (56.5)	5 (14.7)	
13–18	7 (15.2)	9 (26.5)	
19–24	6 (13.0)	-	
≥25	3 (6.5)	11 (32.4)	

**Table 3 ijerph-19-09324-t003:** Overall mental health status, resilience, perceived social support and self-esteem of participants before intervention.

Categories/Level	Frequency (n)/Percentage (%)		
	Mental Health Status		
	**Depression**	**Anxiety**	**Stress**
Normal	26 (32.5)	31 (38.8)	33 (41.3)
Mild	10 (12.5)	15 (18.8)	9 (11.3)
Moderate	19 (23.8)	9 (11.3)	21 (26.3)
Severe	15 (18.8)	8 (10.0)	15 (18.8)
Extremely severe	10 (12.5)	17 (21.3)	2 (2.5)
**Resilience**			
Low	46 (57.5)		
High	34 (42.5)		
**Perceived social support**			
Low	1 (1.3)		
Medium	26 (32.5)		
High	53 (66.3)		
**Self-esteem**			
Low	63 (78.8)		
High	17 (21.3)		
**Coping skills**			
Low	0 (0.0)		
High	80 (100.0)		

**Table 4 ijerph-19-09324-t004:** Baseline mental health status, resilience, perceived social support, self-esteem, and coping skills of participants between groups.

Total Score	Intervention = 46	Control = 34	*p*-Value
**Depression**	7.6 ± 4.3	9.5 ± 4.9	0.271
**Anxiety**	7.7 ± 4.1	5.2 ± 3.1	0.304
**Stress**	10.2 ± 4.2	8.1 ± 4.5	0.518
**Resilience**	113.6 ± 22.3	115.7 ± 25.7	0.154
**Perceived Social Support**	63.4 ± 15.7	62.0 ± 11.8	0.331
**Self-Esteem**	25.6 ± 5.2	26.6 ± 5.2	0.851
**Coping Skills**	50.0 ± 8.9	43.4 ± 9.2	**0.049 ***

**Table 5 ijerph-19-09324-t005:** Correlation analysis between socio-demographic factors and personal factors with dependent variables of participants.

Variables	Pearson Correlation (r)						
	Depression	Anxiety	Stress	Mental Wellbeing	Resilience	Perceived Social Support	Self-Esteem
**Socio-demographic factors**							
Age	−0.222 *	0.044	0.005	-	0.140	0.002	−0.131
Number of siblings	0.737	0.133	−0.176	0.052	0.153	0.194	−0.128
Child order number	0.842	−0.007	−0.047	−0.029	−0.037	−0.077	−0.040
Number of family members in one household	0.261 *	0.258 *	−0.090	0.177	0.122	0.264 *	−0.250 *
Parents’ marital status	−0.118	−0.020	−0.015	−0.054	0.057	−0.005	−0.045
Level of education	−0.105	0.017	−0.045	0.201	0.047	0.047	−0.062
Working experience	0.040	−0.115	−0.104	0.072	−0.056	−0.042	0.002
Household income	0.054	0.039	0.011	0.164	0.042	−0.027	−0.032
**Personal factors**							
Resilience	0.014	0.023	−0.181	-	-	-	-
Perceived social support	−0.036	−0.045	−0.261 *	-	-	-	-
Self-esteem	0.351 **	0.244 **	0.501 **	-	-	-	-

** Significant correlation at *p*-value < 0.01; * Significant correlation at *p*-value < 0.05.

**Table 6 ijerph-19-09324-t006:** A 2-way interaction between time and intervention on mental health status, resilience, perceived social support and self-esteem of participants.

Outcome Parameters	Interaction	Type III Sum of Squares	df	Mean Square	F-Value	*p*-Value	Partial Eta Squared
Depression	Time	87.771	1.718	51.103	0.048	**0.040 ***	0.049
	Intervention	150.553	1.000	150.553	11.985	**0.001 ***	0.158
	Time × Intervention	3.145	1.718	1.831	0.118	0.859	0.002
Anxiety	Time	3.529	1.778	1.764	0.165	0.823	0.003
	Intervention	0.177	1.000	0.177	0.025	0.874	0.000
	Time × Intervention	103.064	1.778	57.959	4.824	**0.012 ***	0.070
Stress	Time	30.783	1.910	16.116	1.631	0.201	0.025
	Intervention	21.528	1.000	21.528	2.799	0.099	0.042
	Time × Intervention	238.116	1.910	124.667	12.617	**<0.001 ***	0.165
Mental wellbeing	Time	440.955	1.698	259.675	3.328	**0.047 ***	0.052
	Intervention	300.191	1.000	300.191	7.354	0.009	0.108
	Time × Intervention	175.241	1.698	103.198	1.323	0.269	0.021
Resilience	Time	190.625	1.787	106.679	0.316	0.705	0.005
	Intervention	1553.674	1.000	1553.674	4.485	**0.038 ***	0.067
	Time × Intervention	1625.167	1.787	1625.167	2.694	0.072	0.042
Perceived social support	Time	13.042	1.989	6.555	0.047	0.953	0.001
	Intervention	413.444	1.000	413.444	4.513	**0.038 ***	0.068
	Time × Intervention	100.167	1.989	50.348	0.362	0.696	0.006
Self-esteem	Time	77.823	1.991	39.079	1.796	0.170	0.028
	Intervention	189.063	1.000	189.063	15.721	**<0.001 ***	0.202
	Time × Intervention	91.219	1.991	45.805	2.105	0.126	0.033
Coping	Time	97.906	1.935	50.610	1.615	0.204	0.025
	Intervention	650.250	1.000	650.250	17.146	**<0.001 ***	0.217
	Time × Intervention	46.781	1.935	24.182	0.772	0.461	0.012

**Table 7 ijerph-19-09324-t007:** A 3-way interaction between time, intervention and sociodemographic factors on anxiety and stress mental health status of participants.

Outcome Parameters	Interaction	Type III Sum of Squares	Df	Mean Square	F-Value	*p*-Value	Partial Eta Squared
Anxiety	Time × Intervention	103.064	1.778	57.959	4.824	**0.012 ***	0.070
	Time × Intervention × Age	30.355	1.000	30.355	1.079	0.303	0.170
	Time × Intervention × Siblings	0.009	1.000	0.009	0.000	0.986	0.000
	Time × Intervention × Child order	0.121	1.000	0.121	0.004	0.948	0.000
	Time × Intervention × Household	3.144	1.000	3.144	0.110	0.741	0.002
	Time × Intervention × Parents	37.330	1.000	37.330	1.332	0.253	0.021
	Time × Intervention × Duration	2.045	1.000	2.045	0.072	0.827	0.001
	Time × Intervention × SocSupport	22.277	1.000	22.277	0.778	0.378	0.013
	Time × Intervention* SelfEsteem	95.880	1.000	95.880	3.543	0.065	0.055
Stress	Time × Intervention	238.116	1.910	124.667	12.617	**<0.001 ***	0.165
	Time × Intervention × Age	0.128	1.000	0.128	0.004	0.952	0.000
	Time × Intervention × Siblings	2.282	1.000	2.282	0.065	0.800	0.001
	Time × Intervention × Child order	10.972	1.000	10.972	0.313	0.578	0.005
	Time × Intervention × Household	45.895	1.000	45.895	1.333	0.253	0.021
	Time × Intervention × Parents	28.732	1.000	28.732	0.828	0.366	0.013
	Time × Intervention × Duration	0.438	1.000	0.438	0.012	0.912	0.000
	Time × Intervention × SocSupport	7.890	1.000	7.890	0.225	0.637	0.004
	Time × Intervention × SelfEsteem	227.677	1.000	227.677	7.240	**0.009 ***	0.106

## Data Availability

No new data were created or analysed in this study. Data sharing is not applicable to this article.

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
