# Peer review of "The Effectiveness of the Super Skills for Life (SSL) Programme in Promoting Mental Wellbeing among Institutionalised Adolescents in Malaysia: An Interventional Study"

_ijerph, 2022, doi:10.3390/ijerph19159324_

Round 1
Reviewer 1 Report
The study examines the effect of an intervention on the mental health and wellbeing of young females in Malaysian institutions, and factors associated with mental health in this group. The authors are to be commended on their work, as paper provides some useful data and insights, and there is limited existing evaluations of such programs in lower- and middle-income countries. However, there are many limitations to the findings around the evaluation of the program, which means that the authors need to be more cautious in interpreting the outcomes. There are also several areas for improvement in the statistical approach.
- Introduction: Please provide a definition of institutionalisation - is the focus only on detention within the justice system (i.e., correctional institutions), or are other forms of institutionalisation also included?
- Section 2.4 doesn't make sense. The outcomes and analyses are all based on continuous variables but the power calculation was based on a difference in proportions. Please clarify. Nonetheless, the level of detail provided in this section is probably unnecessary – just a brief justification is needed.
- The layout of Table 1 could be improved - "Session 1" is missing and the line spacing is inconsistent.
- Although Table 3 is somewhat useful for describing the full sample, it would be more informative if it provided the mean (SD) for each variable, separated by intervention vs control condition. Alternatively, breaking down the categories by intervention vs control (and reporting chi-square statistics) would also be informative.
- The analyses in 3.4 / Table 5 do not directly test the effectiveness of the SSL program. They compare the two groups at each time point, which means they neglect initial levels of each variable. The significant differences at baseline in the outcome measures (and in the background characteristics) indicate that the two groups were not comparable. To show whether the program was effective would require demonstrating that the changes between pretest and post/FU were greater in the intervention group than the control group (this is a 2-way interaction between time and condition). Ideally this could be done using mixed model repeated measures analysis, but given there was little attrition, an alternative would be to use ANOVA, ANCOVA or regression analysis.
- A further important limitation of the analyses is that allocation to the intervention was not random. Therefore, the analyses should adjust for the social and demographic differences between the two conditions. Again, this could be achieved using ANOVA/ANCOVA, regression or MMRM analyses.
- The explanation around increasing number of children as being related to competition would benefit from a reference. It may instead be more related to larger families having fewer resources to support their children (see, e.g., https://doi.org/10.1186/s40359-021-00688-2).
- Based on the shortcomings of the design and analytical approach, the section from lines 266-356 is not sufficiently supported by the analyses. Likewise, the abstract and conclusion and not supported by the analyses. Even with an improved analytical approach, the authors should be more cautious about interpreting the findings due to the study design. It could be the case that conditions in the two institutions were very different (e.g., better programs/services/housing conditions offered at one institution but not the other, different entry criteria, different social backgrounds) such that changes in mental health outcomes could have been unrelated to the SSL intervention - no analysis can fully account for these potential differences.
- The terms "normal" or "regular" suggest that there is something "abnormal" or "irregular" about institutionalised adolescents. Alternative terminology such as "non-insitutionalised adolescents" or "adolescents from the general population" would be preferable and less stigmatising.
- There are some minor wording issues throughout the paper - e.g. "the cases ... had" (the cases have); "Evidences have..." (evidence has...); "although [the] study"; article use - e.g., "SSL programme" should be "the SSL programme"; some problems with tense usage.
Reviewer 2 Report
Introduction section
The background, such as the necessity or importance of the study, was well described.
Methods
Design lines 83-89 would be better described in the data collection section (2.6).
A more detailed description of the instruments used to derive the study results is required. For example, the original tool, who developed it, the composition or the scoring method of the item, and the reliability or validity results should be described.
Results
1) The results on the pre-test homogeneity of the intervention group and the control group (Table 2, Table 3) should be described.
2) Is result as shown in Table 4 necessary for intervention study design?
3) The pre-test, post-test, and f/u results of the intervention group and the control group should be described. It is difficult to think of the results of the intervention.
Discussion should be described according to study results.
Round 2
Reviewer 1 Report
I thank the authors for their comprehensive responsive to my comments. The manuscript is much improved, particularly with respect to the analysis of intervention effectiveness.
Given the mixed efficacy findings (only 2 of 8 outcomes were significant) and the study design (not an RCT), I think the conclusions of the paper (in the abstract and discussion) should be a little more circumspect - e.g., there was evidence that the program had positive effects on stress and anxiety but not on other aspects of mental health.
There are still a few wording issues, particularly with tense use, that could be corrected in copyediting. E.g., "When considered collectively, the mental wellbeing of participants showed a significant improvement as their incarceration duration progresses [progressed]."
Author Response
The authors thank the reviewer for the comments.
- The authors have amended the conclusion in the manuscript according to the reviewer's suggestion. The abstract conclusion however was paraphrased to not be a duplicate of the text conclusion.
- The authors have gone through the manuscript and edited for typography.
Reviewer 2 Report
I have seen the revisions and new descriptions by the researchers. However, please identify the statistical method of lines 191-194 once more.
Author Response
The authors thank the reviewer for the comment. In response to the other reviewer, the authors agreed that the post hoc analysis do not directly test the effectiveness of the SSL program since they compare the two groups at each time point, while neglecting initial levels of each variable. Hence, the authors have removed this analysis entirely. In light to the reviewer comment, the authors have realized that the methodology description for the previous post hoc analysis was not removed. Thus, the author have removed the analysis method description in this round.